# RPLP1 Is Up-Regulated in Human Adenomyosis and Endometrial Adenocarcinoma Epithelial Cells and Is Essential for Cell Survival and Migration In Vitro

**DOI:** 10.3390/ijms24032690

**Published:** 2023-01-31

**Authors:** Riley Peterson, Paige Minchella, Wei Cui, Amanda Graham, Warren B. Nothnick

**Affiliations:** 1Department of Cell Biology and Physiology, University of Kansas Medical Center, Kansas City, KS 66160, USA; 2Department of Pathology and Laboratory Medicine, University of Kansas Medical Center, Kansas City, KS 66160, USA; 3Center for Reproductive Sciences, Institute for Reproductive and Developmental Sciences, University of Kansas Medical Center, Kansas City, KS 66160, USA; 4Department of Obstetrics and Gynecology, University of Kansas Medical Center, Kansas City, KS 66160, USA

**Keywords:** adenomyosis, endometrial adenocarcinoma, endometriosis, RPLP1, cell migration, cell survival

## Abstract

Adenomyosis is defined as the development of endometrial epithelial glands and stroma within the myometrial layer of the uterus. These “ectopic” lesions share many cellular characteristics with endometriotic epithelial cells as well as endometrial adenocarcinoma cells, including enhanced proliferation, migration, invasion and progesterone resistance. We recently reported that the 60S acidic ribosomal protein P1, RPLP1, is up-regulated in endometriotic epithelial cells and lesion tissue where it plays a role in cell survival. To evaluate if a similar pattern of expression and function for RPLP1 exists in adenomyosis and endometrial cancer, we examined RPLP1 expression in adenomyosis and endometrial cancer tissue specimens and assessed its function in vitro using well-characterized cell lines. A total of 12 control endometrial biopsies and 20 eutopic endometrial and matched adenomyosis biopsies as well as 103 endometrial adenocarcinoma biopsies were evaluated for RPLP1 localization by immunohistochemistry. Endometrial adenocarcinoma cell lines, Ishikawa, HEC1A, HEC1B and AN3 were evaluated for RPLP1 protein and transcript expression, while in vitro function was evaluated by knocking down RPLP1 expression and assessing cell survival and migration. RPLP1 protein was up-regulated in eutopic epithelia as well as in adenomyosis lesions compared to eutopic endometria from control subjects. RPLP1 was also significantly up-regulated in endometrial adenocarcinoma tissue. Knockdown of RPLP1 in endometrial adenocarcinoma cell lines was associated with reduced cell survival and migration. RPLP1 expression is up-regulated in eutopic and ectopic adenomyotic epithelia as well as in the epithelia of endometrial cancer specimens. In vitro studies support an essential role for RPLP1 in mediating cell survival and migration, processes which are all involved in pathophysiology associated with both diseases.

## 1. Introduction

The endometrium is the functional lining of the uterus which is essential for successful pregnancy. Abnormalities in the endometrium are thought to contribute to the development of diseases such as endometriosis, adenomyosis, and endometrial cancer. Adenomyosis and endometriosis are benign gynecological conditions common in women of reproductive age in which endometrial-like glands and stroma develop within the uterine smooth muscle/myometrium or outside of the uterine cavity, respectively [1,2,3,4]. Endometrial cancer can be staged according to the International Federation of Gynecology and Obstetrics (FIGO) and American Joint Committee on Cancer (AJCC-tumor, lymph node and metastases) system-TNM [5]. FIGO stage I tumors are limited to the uterine body including growth into the myometrium. Stage II tumors typically invade into cervical stromal connective tissue, but without invading beyond the uterus, while stage III tumors may invade local or regional structures beyond the uterus including the lymph nodes. While adenomyosis and endometriosis are not as aggressive as endometrial cancers, these two pathological tissues share several cellular characteristics with malignant endometrial cancer tissue including invasion, migration and augmented cell survival [6,7]. Further, adenomyosis and endometriosis are often comorbid with one another [8], and both diseases have been associated with endometrial cancer [9]. To date, very few studies have evaluated potential regulators of these cellular processes among the three diseases.

We recently reported over-expression of the 60S acidic ribosomal protein P1 (RPLP1) in endometrial epithelial cells from women with endometriosis [10], while Artero-Castro and colleagues reported over-expression of the RPLP protein family in cancers of the ovary and endometrium [11]. RPLP1 was first identified as a regulator of protein elongation [12]. The *RPLP1* transcript is expressed as a 1174 nucleotide variant (v1; NM_001003) and a 1099 nucleotide variant (v2; NM_213752) which respectively code for the major 12-kDa protein and minor 8.4 kDa protein forms. Emerging data suggest that RPLP1 is a multi-functional protein capable of modulating cell proliferation, migration, invasion and epithelial to mesenchymal transition [13,14,15,16], all of which are common characteristics of epithelial cells of adenomyosis, endometriosis and endometrial cancer. Due to shared physiological characteristics and rates of comorbidity among adenomyosis and endometrial cancer, we set out to further explore the role of RPLP1 in the pathogenesis of adenomyosis and endometrial cancer by assessing its expression in human tissues and cell lines as well as assessing its ability to regulate cell viability and migration in vitro.

## 2. Results

### 2.1. RPLP1 Protein Localization in Adenomyotic Tissue

We first examined RPLP1 expression and localization in uterine tissue from adenomyosis patients and controls (Figure 1) using the 7-point scoring system (0, 0.5, 1.0, 1.5, 2.0, 2.5 or 3.0) defined under Section 4.2. In disease-free controls, RPLP1 protein expression was minimal in both endometrial glands and stroma (H-score = <50) and these levels of expression did not significantly differ (*p* > 0.05) between stages of the menstrual cycle (Figure 1A,B, upper panel and left and right bar graphs). In contrast, eutopic endometria from subjects with adenomyosis expressed significantly (*p* < 0.05) higher levels of RPLP1 in endometrial glands during the proliferative stage of the menstrual cycle based on H-Score assessment (Figure 1C upper panel and left and right bar graphs), but not during the secretory stage (Figure 1D, upper panel and left and right bar graphs). Stromal cell expression of RPLP1 in these same samples was minimal (H-score = <50; Figure 1C,D, upper panel and left and right bar graphs), much like that in control specimens (Figure 1A,B). Similar to adenomyotic eutopic endometria during the proliferative stage (Figure 1C), adenomyotic lesions (foci; Figure 1E upper panel and left and right bar graphs) exhibited significantly (*p* < 0.05) higher levels of glandular RPLP1 expression compared to controls (Figure 1A). This level of epithelial cell expression did not differ compared to eutopic endometrial epithelial cell expression from matched biopsies (Figure 1E vs. Figure 1C). Adenomyotic foci also exhibited significantly (*p* < 0.05) higher levels of RPLP1 expression in secretory stage biopsies (Figure 1F, upper panel and left and right bar graphs) compared to matched eutopic tissue (Figure 1D) and eutopic tissue from control subjects (Figure 1B). Ectopic adenomyotic stromal tissue expression of RPLP1 was minimal (H-score = <50), similar to eutopic endometrial stromal from adenomyosis subjects as well as controls. Comparisons between menstrual cycle stages within tissue type revealed that in eutopic endometria from adenomyosis subjects, RPLP1 expression was significantly (*p* < 0.05) lower during the secretory stage of the menstrual cycle compared to the proliferative stage. Interestingly, this cycle-stage pattern of RPLP1 expression was lost in adenomyotic foci as RPLP1 levels remained high in both groups (Figure 1E,F, upper panel and left and right bar graphs).

### 2.2. RPLP1 Protein Localizatoin in Endometrial Carcinoma

We next examined RPLP1 expression in endometrial adenocarcinomas compared to control endometrial tissue. RPLP1 expression was evaluated in a human endometrial carcinoma tissue array from patients who ranged in age from 40 to 76 years. In all cancer biopsies, RPLP1 localized to epithelial cells (100%) and not within surrounding tissue (Figure 2), while samples from control subjects exhibited low to absent staining for RPLP1 in endometrial glands (Ge) and stroma (St). Semi-quantitative assessment (H-score calculation) revealed that compared to non-cancer controls, RPLP1 expression was significantly (*p* < 0.05) higher in all stages of cancer, but there was no difference among the mean level of RPLP1 expression among the three stages of endometrial carcinoma.

Due to the fact that the average age of the subjects in endometrial cancer stages I–III was significantly (*p* < 0.05) higher compared to controls (Figure 3), we wanted to confirm that the elevated expression in these samples was not due to age. To do so, we separately analyzed RPLP1 H-scores from age-matched Stage II and III subjects with controls (grouping proliferative and secretory controls, as their H-scores did not differ statistically; *p* > 0.05). As depicted in Figure 4, RPLP1 was higher in Stage II and III specimens from subjects of similar ages compared to those of controls, verifying that elevated RPLP1 is associated with disease pathology and not the advanced age of these subjects as a whole.

### 2.3. RPLP1 Expression in Endometrial Adenocarcinoma Cell Lines

To begin to evaluate the functional consequence of elevated RPLP1 expression in endometrial cancer, we implemented well-characterized cell lines which represent Type I human endometrial adenocarcinomas (Ishikawa) and Type II human endometrial carcinomas (HEC-1A, HEC-1B and AN3 CA; [17]). We first assessed the level of RPLP1 transcript and protein expression in each cell type (Figure 5). Both variants of *RPLP1* were detected in all four cell lines plus the endometriotic epithelial cell line, 12Z, which was used as a positive control. *RPLP1* variant 1 (V1) was the more abundant transcript (approximately 6 to 7 fold higher) compared to variant 2 (V2), with both Ishikawa and AN3 cells exhibiting the highest levels of expression (expressed as RPLP1 delta ct value—β-actin delta ct value). The relative level of *RPLP1* V1 and V2 was calculated using the delta-delta ct method and is reported as a fold change from the HEC-1A cells (Table 1) which expressed the lowest levels of transcripts/highest RPLP1—β-actin levels (Figure 5, lower panel).

RPLP1 protein was also detected in all four endometrial adenocarcinoma cell lines as well as the 12Z cell line, with AN3 CA (AN3) cells exhibiting the lowest level of protein expression (Figure 5, lower panel) despite the highest level of *RPLP1* transcript expression (Figure 5, upper panel).

### 2.4. RPLP1 Regulates Endometrial Adenocarcinoma Cell Survival and Migration

As we previously reported that RPLP1 was essential for in vitro cell survival of the endometriotic epithelial 12Z cell line, we first determined if a similar pro-survival function existed in the endometrial adenocarcinoma cells. After 48 h, NT-siRNA transfected HEC-1B cells exhibited the highest cell viability index followed by Ishikawa, HEC-1A and AN3 cells (Figure 6). A similar pattern of survival was detected in all cell lines transfected with RPLP1 siRNA; however, when compared to the NT siRNA-transfected cells within each cell type, RPLP1 knockdown was associated with a significant reduction in cell viability, most notably in HEC-1B cells (Figure 6).

To evaluate if RPLP1 expression is associated with cell migratory ability, we conducted scratch assays using NT- and RPLP1-siRNA-transfected Ishikawa, HEC-1A, HEC-1B and AN3 cells and compared gap closure among groups. As depicted in Figure 7, at 24 h, Ishikawa, HEC1-A and HEC1-B cells showed similar rates of migration with the percentage of wounded/scratched area being between 20 and 25%. In contrast, AN3 cells showed significantly lower levels of migration (less than 10%) compared to the other three cell types. Knockdown of RPLP1 (RPLP1 siRNA) was effective in Ishikawa, HEC1-A and HEC1-B, resulting in migration values ranging between 7.5 to 10%; meanwhile, AN3 cells also showed low levels of migration in response to RPLP1 knockdown (on average 8%), which did not differ among cell types. When compared within cell types, all three cells except the AN3 cells showed significant differences between NT- and RPLP1 siRNA-transfected cells (Figure 7).

## 3. Discussion

Endometriosis, adenomyosis and endometrial cancer share many cellular attributes in which the endometrial epithelial (and stromal cells in endometriosis and adenomyosis) exhibit enhanced cell survival and migration, which are hallmarks of all three diseases. Unsurprisingly, RPLP1 has emerged as a novel factor in the pathophysiology of these diseases as well as other cancers outside of the female reproductive tract, which include liver [13] and breast cancer [15]. RPLP1 was first reported to modulate cell proliferation by Artero-Castro and colleagues [18] who demonstrated that over-expression of RPLP1 was able to bypass replicative senescence in primary mouse embryonic fibroblasts by activating the E2F1 promoter. Subsequently, RPLP1 has been reported to be essential for embryonic development of the nervous system [19], modulation of autophagy [20], cell cycle progression [20], and facilitation of viral protein synthesis of Zika virus, dengue viruses and yellow fever virus [21]. These reports support a multifunctional role for RPLP1 beyond its classical regulation of protein elongation.

RPLP1 is an intriguing protein in that it is necessary for the normal physiological and developmental processes described above, and its over-expression is associated with the pathologies described earlier. Several studies have evaluated RPLP1 expression at the genomic/transcript level. Ishii and associates reported that the *RPLP1* gene is highly expressed in keratinocytes, skin and lymph node dissection [22]. Additional studies support not only the robust levels of *RPLP1* transcript, but also consistency in its levels of expression across diverse tissue types, including whole Atlantic cod [23], rat hippocampus from both control and pilocarpine-induced mesial temporal lobe epilepsy [24], human sperm, [25] and rat livers from both normal and insulin-resistant rats [26], among others. In addition, *RPLP1* mRNA expression has been reported to remain stable under hypoxic conditions in neuronal stem cells [27] and chronically hypoxic rat hearts [28]. However, *RPLP1*, along with other ribosomal genes, displayed a rising pattern with increasing duration of reprogramming in induced pluripotent stem cells (iPS; [29]), an observation which is in line with the concept of iPS reprogramming, wherein the change in pluripotency state from less pluripotent to more pluripotent is associated with an increase in cell cycle progression [30,31,32].

*RPLP1* mRNA appears to be abundant, and under most circumstances or experimental conditions may remain relatively consistent across parameters. While most studies have assessed only *RPLP1* transcript levels, very few have assessed both RPLP1 mRNA and protein in the same tissue. Our prior observation [10] that *RPLP1* transcript expression is robust in eutopic endometrial tissue and endometriotic lesion tissue (with only the latter exhibiting significantly higher levels of protein expression) may suggest complex regulation of protein expression, which may include post-transcriptional regulation. The level or abundance of a given mRNA is assumed to be proxy for transcriptional activity and subsequent protein translation. However, transcriptional and post-transcriptional regulatory processes are complex [33], and include multiple factors such as microRNAs [34]. To date, little to no information exists on potential post-transcriptional regulation of RPLP1. Given the findings from our prior study [10] and the results reported in this study, assessment of post-transcriptional regulation of RPLP1 protein expression will provide a deeper understanding of the physiologic and pathophysiologic role of this multifunctional protein in uterine tissue.

In contrast to post-transcriptional regulation, transcription of the RPLP1 gene is more well understood. Predicted transcription factors that bind to the RPLP1 promoter and induce transcription include Nfe2-related transcription factor (NRF) and activator protein-1 (AP-1) [22]. AP-1 regulates c-Myc expression which is elevated in endometrial cancer [35] and endometriosis [36,37]. We had previously demonstrated via Myc knockdown that in the endometriotic epithelial cell line, 12Z, Myc is also associated with induction of RPLP1 protein expression [10]. In contrast to endometriosis and endometrial cancer, c-Myc expression in adenomyosis is less known. A single report by Milde-Langosch and colleagues [38] reported that approximately 25–30% of cervical carcinomas and severe dysplasias exhibited over-expression of c-Myc, but none of the evaluated endometrial lesions that included adenomyosis specimens exhibited c-Myc over-expression. Additional factors that are known to regulate RPLP1 include the environmental toxicant, 2,3,7,8-tetrachlorodibenzo-p-dioxin (TCDD). Jin and coworkers [39] reported that TCDD induced RPLP1 expression in normal human bronchial epithelial cells. It is well established that exposure to this compound is thought to play a role in endometriosis pathogenesis; this is supported by both human [40,41] and animal model studies [42]. With respect to endometrial cancer, an in vitro study using endometrial epithelial cell lines supported the notion that TCDD and its related compounds might create an additional burden of carcinogenicity in estrogen target tissue [43], while findings from epidemiological studies support an increased overall risk for all cancers and increased risk of breast and endometrial cancer from dioxin exposure in adult females [44,45,46].

Our findings that RPLP1 supports both cell survival and migration in vitro are in accordance with the findings of others [13,14,15], and may arise by affecting epithelial to mesenchymal transition. Overexpression of RPLP1 in MDA-MB-231 breast cancer cells was associated with significant increases in protein expression for N-cadherin, snail and vimentin. Significant reduction in E-cadherin and increased metastases in vivo, while knockdown of RPLP1 using shRNA reversed these outcomes [15]. As such, it seems plausible that RPLP1 may modulate cell migration in endometrial adenocarcinoma cell lines via a similar impact on epithelial to mesenchymal transition. Of note were the low levels of cell migration in AN3 cells regardless of if RPLP1 was knocked down or not. These cells also exhibited the lowest levels of proliferation and RPLP1 protein expression, which may suggest that RPLP1 has a lesser role in cell survival/migration in these cells compared to Ishikawa, HEC1-A and HEC1-B. AN3, HEC1A and HEC1B are considered type II endometrial adenocarcinoma cell lines [17]. HEC1-A is a sub-strain of HEC1-B which was isolated from a patient with Stage IA endometrial cancer, while AN3 cells were established from lymph node metastasis resected from a patient with an adenocarcinoma presenting with acanthosis nigricans. According to FIGO classification [5], AN3 cells would be derived from a stage III cancer compared to the more confined stage IA endometrial cancer (HEC1-A and HEC1-B cell lines). Whether the stage of cancer from which the cell lines were established has an impact on RPLP1 function or necessity for survival and/or migration remains to be determined, but could possibly explain the differences observed in this study.

In addition to playing a functional role in adenomyosis, endometrial cancer and endometriosis pathophysiology, RPLP1 may have value as a diagnostic marker. The gold standard for endometriosis diagnosis is laparoscopy [47] with subsequent histological assessment of suspected lesion tissue. Adenomyosis can be diagnosed by a combination of clinical history, gynecological examination, transvaginal ultrasound and magnetic resonance imaging [48], while endometrial cancer is diagnosed by endometrial biopsy and histological confirmation [49]. Outcomes from the current study indicate that for adenomyosis, RPLP1 expression is increased in eutopic endometrial tissue, and our prior results [10] suggest that it is also expressed in eutopic endometria from subjects with endometriosis. It seems plausible that endometrial biopsy and staining intensity for RPLP1 may serve as a less invasive means for diagnosing these two benign uterine diseases. In our current study, regardless of patient age, RPLP1 was consistently overexpressed in adenomyosis and endometrial cancer tissue. The use of RPLP1 in immunohistochemistry screening of endometrial tissue biopsies can assist in identification of disease tissue. With the current lack of biomarkers used for identification of adenomyosis, endometriosis and endometrial cancer, RPLP1 could prove to be a less invasive biomarker. More efficient screening for disease would allow for faster diagnosis and a better chance for therapeutic intervention and patient outcomes. However, larger studies would have to be conducted to evaluate the ability of RPLP1 endometrial tissue expression to distinguish between controls and subjects who may have either endometriosis and/or adenomyosis.

While the potential application of RPLP1 as a less invasive diagnostic marker may be viewed as a strength of the study, additional strengths include the use of multiple, well-defined cell line models, and assessment of both RPLP1 mRNA and protein expression in these cell lines, coupled with assessment of disease tissue from well-defined patient populations. In addition to these strengths, we acknowledge there are also limitations. Our in vitro findings are based upon two-dimensional assay systems which do not fully reflect the in vivo microenvironment which may influence cell behavior. Future studies could incorporate three-dimensional (3D) assays to confirm using extracellular and tumor microenvironment matrices to mimic in vivo tumor conditions, similar to the study reported by Apu and colleagues [50]. Lastly, to further enrich our understanding of a tumor microenvironment in vitro, we could expose primary endometrial epithelial cells to environmental toxins that cause cancer, such as TCDD, to see if we could induce RPLP1 expression similar to that observed in the cancer cell lines used in this study and correlate this expression with that of other markers of proliferation and EMT.

In summary, RPLP1 protein is over-expressed in adenomyosis and endometrial cancer compared to endometrial tissue from women without disease. Two-dimensional (2D) in vitro studies support a functional role of RPLP1 in both cell survival and migration, both processes which are hallmarks of adenomyosis and endometrial cancer. Elevated levels of RPLP1 appear to be a common phenomenon in these diseases as well as endometriosis, further supporting the notion of common pathways which may exist in all three diseases. What remains to be determined is which mechanisms lead to this elevated expression, and if these mechanisms are similar or diverse among subjects with adenomyosis, endometrial cancer and endometriosis.

## 4. Materials and Methods

### 4.1. Human Subjects and Tissue Acquisition

The study was approved by the institutional review board of the University of Kansas Medical Center. Written informed consent was obtained prior to surgery. Controls consisted of patients without evidence of adenomyosis or endometrial carcinoma (N = 12). Full uterine cross sections were obtained from patients diagnosed with adenomyosis (N = 20), and both eutopic and adenomyotic foci were evaluated in the same specimen. Endometrial cancer specimens were obtained as a tissue array (TissueArray.Com LLC, Derwood, MD, USA) which contained punch biopsies from Stage I (N = 22), Stage II (N = 52) and Stage III (N = 29). All tissues (control, adenomyosis, and endometrial carcinoma punch biopsies) were obtained from formalin-fixed, paraffin-embedded blocks which were fixed and processed in similar fashion. For control and adenomyosis specimens, the stage of menstrual cycle was determined by the date of the subject’s last menstrual period for pre-menopausal subjects.

### 4.2. Immunohistochemistry Staining and Quantitation

Archived tissues were obtained from the Department of Pathology and Laboratory Medicine at the University of Kansas Medical Center for adenomyosis and control patients. These tissues, as well as the tissue array slides containing endometrial carcinoma specimens (defined above), were fixed with 10% neutral buffered formalin and subjected to antigen retrieval. The slides were then dehydrated and rehydrated following standard procedures and subjected to immunohistochemical (IHC) localization using RPLP1 (Anti-RPLP1 antibody; Abcam 121190; Abcam, Cambridge, MA, USA) at a dilution of 1:300) as previously described [10]. IHC was performed following the recommendations of the manufacturer using a VectaStain ABC system (Vector Laboratories, Inc., Burlingame, CA, USA), and sections were counterstained with hematoxylin (Vector Laboratories, H-3404-100). Protein localization was identified as dark brown coloring on the tissue slides. Positive controls consisted of human endometriotic lesion tissue [10], while negative controls consisted of slides in which the primary antibody was omitted and replaced with an isotype-matched antibody as well as negative control tissue (myometrium).

To semi-quantitatively assess immunohistochemically detected RPLP1 protein expression, we used the H-score system in which the level of protein was quantified in each cell type (epithelial, stroma, etc.) indicated as a region of interest (ROI). We randomly selected three sections as ROI within each of the endometrial sections of whole uterine cross sections (eutopic control and eutopic adenomyosis) and took average scores from those three sections as the final score. Due to the limited amount of tissue in adenomyotic foci and endometrial carcinoma punch biopsies, we quantified the entire cross section as the ROI. The reaction product signal intensity was then scored using a standard staining chart generated in our laboratory (Figure 8) with the following scores: 0 (absent), 0.5–1 (minimal), 1.5–2 (moderate) or 2.5–3 (strong). The 0.5, 1.5 and 2.5 values were assigned to those specimens that were scored between two categories (for example, a score of 1.5 was assigned to samples that expressed a signal in between that of a 1.0 and 2.0) (Figure 8).

The percentage of cells expressing each level of staining intensity was calculated in each ROI. For all cells of epithelial origin in all specimens, the percentage of cells expressing RPLP1 was 100%, and this localization was cytoplasmic. The H-score was then calculated using the average level of intensity (0, 0.5, 1.0, 1.5, 2.0, 2.5 or 3.0) multiplied by the percentage of cells at that intensity to generate the semi-quantitative measurement of the positive staining.

### 4.3. mRNA Isolation and qRT-PCR

Quantitative real-time PCR (qRT–PCR) was performed as previously described [10]. Briefly, total RNA was isolated using Tri-Reagent (Sigma Chemical Co., St. Louis, MO, USA) according to recommendations of the manufacturer. Total RNA (1 µg in 20 µL) was reverse transcribed using reverse transcription (RT) kits (Applied Biosystems; Foster City, CA, USA) following the manufacturer’s protocol. Primers for human *RPLP1* variants 1(v1) and 2 (v2) as well as beta-actin (*ACTB*) were designed using Primer-Blast and synthesized by Integrated DNA Technology (IDT, Coralville, IA, USA): human *RPLP1v1* (NM_001003): forward, 5′-TGACAGTCACGGAGGATAAGA-3′ and reverse, 5′-CCAGGCCAAAAAGGCTCAAC-3′; human *RPLP1v2* (NM_213725): forward, 5′-CTCACTTCATCCGGCGACTA-3′ and reverse, 5′-GCCAGGGCCGTGACTGT-3′; human *ACTB* (NM_001101): forward 5′-GCACAGAGCCTCGCCTTT-3′; reverse, 5′-TATCATCATCCATGGTGAGCTGG-3′. The resulting material was then used for independent qRT–PCR. qRT–PCR was carried out on an Applied Biosystems HT7900 Sequence Detector. To account for differences in starting material, ACTB primers were used. All samples were run in triplicate and the average value was used in subsequent calculations. The 2-delta-delta CT method was used to calculate the fold-change values among samples as previously described [10].

### 4.4. Protein Isolation and Western Blot Analysis

Total protein was extracted from each of the cell lines (non-transfected, NT or RPLP1 siRNA reverse transfected plated at a density of 3.0 × 10^5^ cells/well) using RIPA buffer (1X RIPA, Catalog #9806, Cell Signaling Technologies [CST], Danvers, MA, USA). The protein concentration in each sample was determined using the Bio-Rad Protein Assay ([Catalog 3500-0006], Bio-Rad Laboratories, Richmond, CA, USA). The same amount of protein (30 μg) was subjected to 12% bis(2-hydroxyethyl)amino-tris(hydroxymethyl)methane (*w*/*v*) gel electrophoresis and electroblotted onto PVDF membranes (Invitrogen). RPLP1 (21636-1-AP; 1:400; Proteintech, Rosemont, IL, USA) and donkey, anti-rabbit secondary antibody (catalog #NA934V; 1:20,000; GE Healthcare/Fisher Scientific, Pittsburgh, PA, USA) were used. Stripping and re-probing for β-actin (ab8227; 1:10,000; Abcam, Cambridge, MA, USA) was conducted to normalize RPLP1 protein expression levels.

### 4.5. Cell Culture and Transfection

The endometriotic epithelial cell line, 12Z, was obtained from Dr. Linda Griffith (Massachusetts Institute of Technology, Cambridge, MA, USA), while the Ishikawa cell line (Type I endometrial carcinoma; early stage disease [17]) was provided by Dr. Bruce Lessey (Wake Forest University School of Medicine, Winston-Salem, NC, USA). HEC-1-A, (ATCC^®^ HTB112™), HEC-1-B (ATCC^®^ HTB-113™), and AN3 CA (ATCC^®^ HTB-111™) cell lines (Type II endometrial carcinoma; advanced stage disease [17]) were purchased from the American Type Culture Collection (ATCC; Manassas, VA, USA). Cell culture was conducted following the general approach as previously described [10]. Briefly, cells were cultured in phenol red-free Dulbecco’s Minimum essential medium (DMEM)/Ham’s F12 (Fisher Scientific, Pittsburgh, PA, USA) + 10% charcoal stripped FBS (Atlanta Biologicals, Atlanta, GA, USA) containing penicillin-streptomycin (Pen-Strep; 100 units/100 µg mL, respectively; Life Technologies, Carlsbad, CA, USA) and normocin (100 µg/mL; InvivoGen, San Diego, CA, USA; Catalog # ant-nr) in T75 flasks until approximately 90% confluency. Cells were then passed and prepared for reverse transfection as follows. Each cell line was separately transfected with siRNA to RPLP1 (ON-TARGETplus SMARTPool; L-011135-00; GE Healthcare Dharmacon, Inc.; Lafayette, CO, USA) or a non-targeting (NT) mimic (siGENOME Non-targeting Control siRNA #2; D-001210-02; GE Healthcare Dharmacon, Inc.) which was used as a negative control (50 nM final concentration of each). To do so, siRNAs were individually incubated with siPORT™ *NeoFX*™ transfection agent (ThermoFisher) in DMEM:F12 media containing Pen-Strep for 10 min. To these mixtures, each cell line was added in an equal volume of DMEM:F12 media containing Pen-Strep plus 2% (*v*/*v*) FBS to bring the final concentration to 62,500 cells/mL media, which was then diluted for each experimental endpoint as defined under each experimental endpoint as follows. Reverse transfected cells were then plated in 6-well (final volume 2.5 mL media/well) plates (Avantor/VWR Scientific; Radnor, PA, USA) as described for each assay in the following sections. Due to the high expression levels of RPLP1, a second transfection was conducted 24 h after the initial reverse transfection. Media was removed from each well and DMEM:F12 plus P/S and 2% FBS was added followed by transfection with lipofectamine-2000 and NT or RPLP1 siRNA (50 nM final concentration) as previously described [10]. Twenty-four hours after the second transfection, the impact of RPLP1 depletion on cell viability and migration was assessed as follows.

### 4.6. Cell Viability/Crystal Violet Staining Assays

Each cell type was double transfected as described above with either NT or RPLP1 siRNA and plated individually into 6-well plates at a density of 150,000 cells/well. Cells were cultured for 48 h, after which cell survival was quantified using crystal violet staining [51,52]. Briefly, spent media were removed by aspiration and each well was rinsed once with PBS, followed by addition of 1 mL of staining solution (containing 0.125 g crystal violet powder in 50 mL of 20% (*v*/*v*) methanol) for 10 min at 22 °C. The staining solution was then removed, and cells were rinsed 6 times with 2 mL PBS followed by addition of lysis buffer (0.1 M sodium citrate, 50% ethanol [*v*/*v*], pH 4.2). Cells were lysed for 30 min at 22 °C on a rocking platform, after which lysate was diluted 1:20 with ddH2O and read at 590 nm on a spectrometer. The data were expressed as OD values for NT-siRNA and RPLP1-siRNA transfected cells and were assayed in duplicates.

### 4.7. Cell Migration/Scratch Assays

Cell migration ability was examined using scratch assays [13] with minor modifications. Briefly, a 0.5 cm scratch/wounded area was marked on the bottom of the 6-well plates on which double-transfected NT- and RPLP1 Ishikawa, HEC1-A, HEC-1B and AN3 were individually seeded into each well and cultured to near confluency. At that point, the wound was created by scraping the cells within the indicated marked area using a pipet tip with a diameter of 0.5 cm (1000 µL pipet tip cut to 0.5 cm diameter). Cells were then cultured in serum-free DMEM:F12 media with P/S, with the initial gap width (0.5 cm) assessed under an inverted microscope at 100× at time 0 h. The percentage cell migration was that determined after 24 h using the following formula: ([scratch width at 0 h) − (scratch width at 24 h)]/[scratch width at 0 h]) × 100%, and is reported as the percent cell migration.

### 4.8. Statistical Analysis

Statistical analysis was conducted using GrapPad Instat3.10 (San Diego, CA, USA). Human IHC data were first evaluated for normalcy of distribution using a Kolmogorov and Smirnov test, and did not follow Gaussian distributions. As such, comparisons among the study groups were then made using the non-parametric one-way ANOVA equivalent Kruskal–Wallis test or the non-parametric *t*-test, the Mann–Whitney U-test, as indicated in the figure legends. For cell line studies, comparisons among the different cell types for each endpoint were made using a one-way ANOVA, while comparisons between NT- and RPLP1-siRNA-transfected cells were made using unpaired *t*-tests. For one-way ANOVA data, when an F test indicated statistical significance, post hoc analysis was carried out using the Tukey HSD procedure. Significance was set at *p* < 0.05 for all experiments.

## Figures and Tables

**Figure 1 ijms-24-02690-f001:**
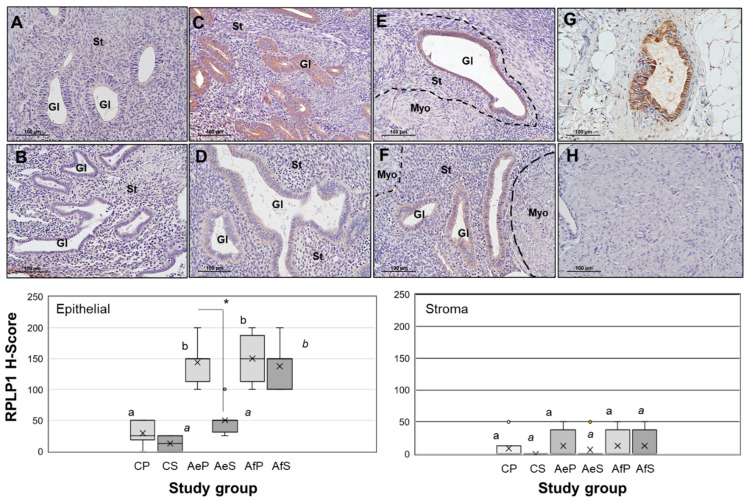
Immunohistochemical localization of RPLP1 in control and adenomyotic tissue. Upper panel: RPLP1 protein was localized in endometrial/uterine tissue as described under Materials and Methods, and images are representative of six control specimens from the proliferative ((**A**); N = 6) and secretory ((**B**); N = 6) stages of the menstrual cycle, while eight matched eutopic and adenomyotic foci were evaluated from subjects with adenomyosis during the proliferative ((**C**) = eutopic, (**E**) = adenomyosis; N = 8) and secretory ((**D**) = eutopic, (**F**) = adenomyosis; N = 8) stages of the menstrual cycle. St = stroma, Gl = glands, Myo = myometrium. Scale bar = 100 µm, and magnification is at 200× for all images. The broken lines in (**E**,**F**) indicate demarcation between the myometrium and endometrium (glands and stroma) within the myometrial layer. (**G**) = positive control endometriotic lesion tissue, (**H**) = negative control myometrium. Lower left (glandular epithelial cells) and right (stromal cells) bar graphs: H-Scores for RPLP1 staining in control and adenomyosis samples. CP = control—proliferative (N = 6); CS = control—secretory (N = 6); AeP = adenomyosis—eutopic—proliferative (N = 8); AeS = adenomyosis—eutopic—secretory (N = 8); AfP = adenomyosis—foci—proliferative (N = 8); and AfS = adenomyosis—foci—secretory (N = 8). Different letters indicate statistical significance among the means within proliferative samples (non-bold letters) and secretory samples among the three tissue type study groups (italic letters). * indicates statistical significance (*p* < 0.05) between the means within eutopic endometria from adenomyosis subjects between the proliferative and secretory stages. X indicates the mean value in each study group. There were no significant differences between mean H-score for CP vs. CS or AfP vs. AfS (*p* > 0.05).

**Figure 2 ijms-24-02690-f002:**
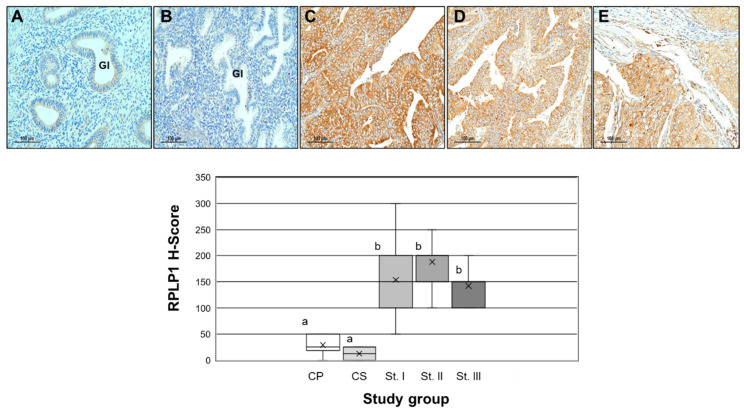
RPLP1 localization in endometrial cancer. Upper panel: Endometrial biopsy tissue from control proliferative endometrial biopsy ((**A**); N = 6), control secretory endometrial biopsy ((**B**); N = 6), Stage I endometrial carcinoma ((**C**); N = 21), Stage II endometrial carcinoma ((**D**); N = 36) and Stage III endometrial carcinoma ((**E**); N = 26) were prepared for immunohistochemical localization of RPLP1. Scale bar = 100 µm; Gl = endometrial glands. Lower panel: H-score quantitation for RPLP1 among the study groups. CP = control proliferative (N = 6); CS = control secretory (N = 6); St I—endometrial carcinoma Stage I (N = 21); St-II = endometrial carcinoma Stage II (N = 36); St-III = endometrial carcinoma Stage III (N = 26). Different letters indicate statistical significance among the means by one-way ANOVA. X represents the mean in each study group.

**Figure 3 ijms-24-02690-f003:**
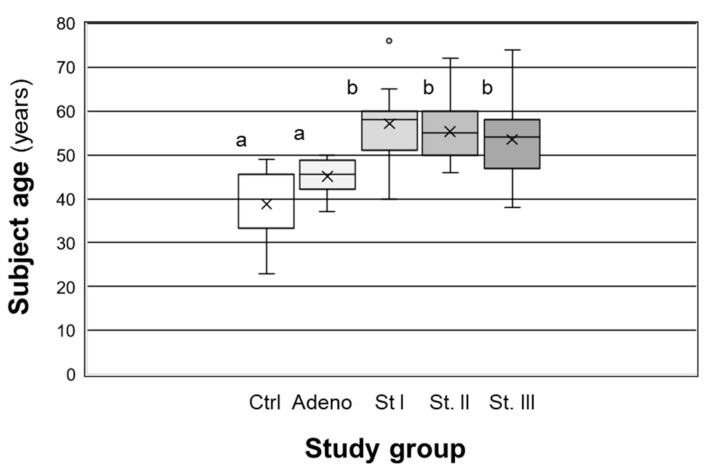
Age range for subjects in each of the study groups. Data on patient age were examined among the study groups by one-way ANOVA for control (Ctrl), adenomyosis (Adeno) and endometrial carcinoma stages I (St I), II (St II) and III (St III). Different letters indicate statistical significance among the means. X represents the means in each study group, while the circle indicates suspected outiers.

**Figure 4 ijms-24-02690-f004:**
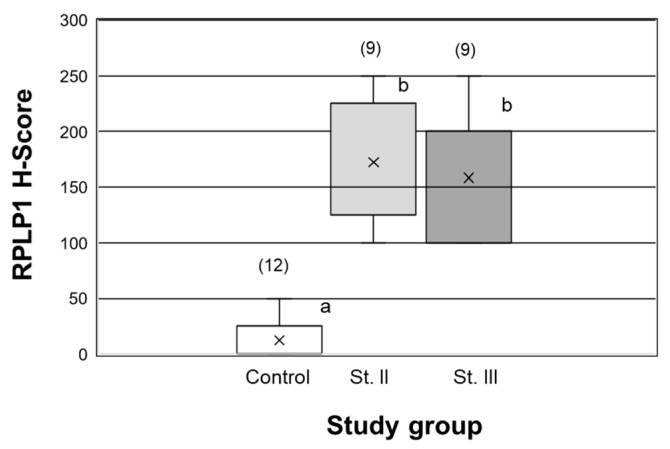
RPLP1 expression in endometrial tissue across study groups matched for age. H-scores are depicted for age-matched controls (Control) and age-matched Stage II (St II) and Stage III (St III) cancer subjects. Different letters indicate statistical significance among the means, and the numbers above each bar represent the sample size (N) assessed in each group (*p* < 0.001). Stage I samples were not age matched compared to controls because the age range was similar between groups (*p* > 0.05). X represents the means in each study group.

**Figure 5 ijms-24-02690-f005:**
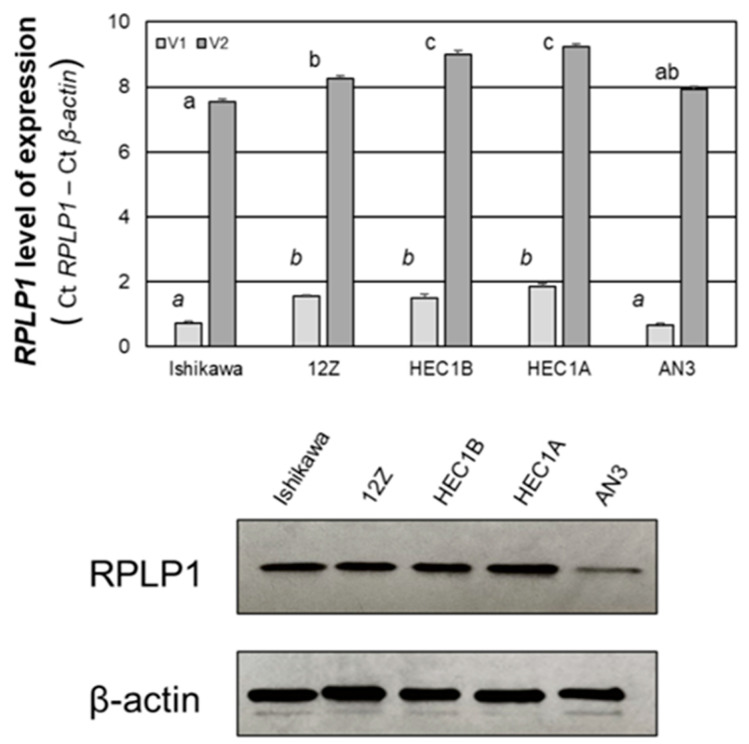
RPLP1 transcript and protein expression in endometrial adenocarcinoma cell lines. RNA and protein were isolated from each cell line. RPLP1 transcript expression (upper panel, variants 1 [V1] and 2 [V2]) and protein (lower panel) were assessed by qRT-PCR and Western blot, respectively. For transcript levels, different letters indicate statistical significance among the mean values as determined by one-way ANOVA for each transcript variant, with block letters for V2 and italics for V1 (N = 4). Western blot data are representative of two independent sample sets (N = 2).

**Figure 6 ijms-24-02690-f006:**
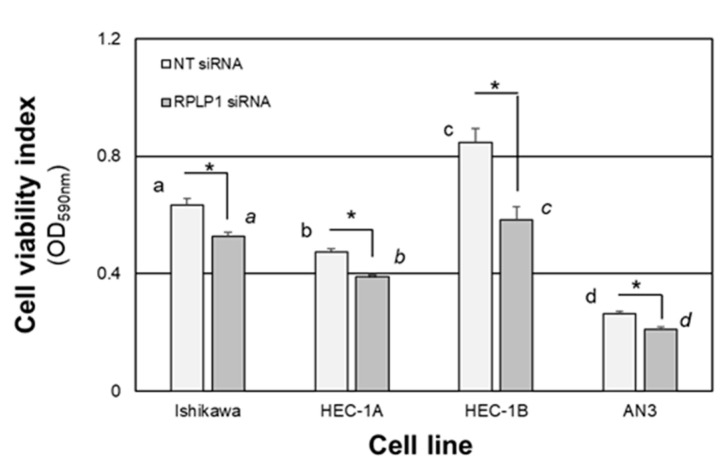
RPLP1 knockdown impairs cell survival. Endometrial adenocarcinoma cell lines were transfected with a non-targeting (NT) or RPLP1 siRNA and cultured for 48 h, after which cell viability was determined by crystal violet staining. Data are expressed as the average OD at 590 nm for each cell line and treatment group ± SEM (N = 4). Different letters indicate statistical significance among the means within transfection groups across the different cell lines, with block letters for NT siRNA-transfected cells and italicized letters comparing means for the RPLP1 siRNA-transfected cells using a one-way ANOVA within each treatment group. Asterisks indicate statistical significnace between transfection groups within each cell type using unpaired *t*-tests.

**Figure 7 ijms-24-02690-f007:**
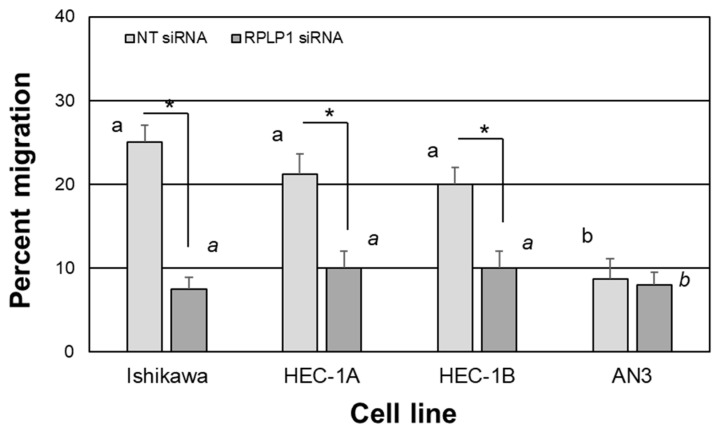
RPLP1 knockdown impairs cell wound recovery/cell migration. Wound/scratch assays were conducted as described under “Materials and Methods”. Data are expressed as the percent recovered wounded area *±* SEM and are reflective of the cell’s ability to migrate into the scratch area (N = 4). Different letters indicate statistical significance among the means within transfection groups across the different cell lines, with block letters for NT siRNA-transfected cells and italicized letters comparing means for the RPLP1 siRNA-transfected cells using a one-way ANOVA within each treatment group. Asterisks indicate statistical significance between transfection groups within each cell type using unpaired *t*-tests.

**Figure 8 ijms-24-02690-f008:**
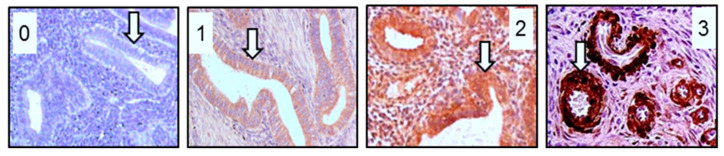
Representative H-score immunohistochemistry (IHC) scoring system. Tissue sections from endometria were stained and assigned a score of 0 = absent/no staining; 1.0 = minimal staining, 2 = moderate staining or 3.0 = strong staining. For staining intensity between integers, 0.5 units were added if evaluators felt staining was mid-way between the two whole integer intensity levels. The white arrow indicates the intensity level in each tissue section used as the reference score. Magnification was at 200×.

**Table 1 ijms-24-02690-t001:** Relative level of RPLP1 V1 and V2 transcript among the cell lines.

Cell Line	Fold Change V1	Fold Change V2
HEC-1A	1	1
Ishikawa	3.23	3.21
12Z	1.85	1.96
HEC-1B	1.88	1.16
AN3	3.37	2.46

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
