# Peer review of "RPLP1 Is Up-Regulated in Human Adenomyosis and Endometrial Adenocarcinoma Epithelial Cells and Is Essential for Cell Survival and Migration In Vitro"

_ijms, 2023, doi:10.3390/ijms24032690_

Round 1

Reviewer 1 Report

Please see PDF file attached.

Author Response

ijms-2164710_Response to reviewers

     We would like to thank the reviewers for their helpful insights.  We have answered each of the  queries as outlined below (response is underlined).  We also indicate where the changes appear in the revised manuscript herein as well as in the marked-up version of the revised manuscript.

Reviewer 1

The paper by Peterson/Minchella et al. reports an upregulation and over-expression of RPLP1 in human tissue of adenomyosis, endometriosis and endometrial carcinoma, which they relate to an important role of this protein in cell migration and survival in these diseases. Their results are certainly interesting and add these lesions to an already known number of diseases with increased RPLP1 activity. Yet there are several points to be clarified and the manuscript has to be revised in a number of ways:

  1. In the abstract, the authors describe RPLP1 as a "novel protein" they have identified. This is not correct, since according to their own citations (e.g. [11] and [12]) this protein has already been described over 20 years ago and its role in normal and pathologic tissue has been studied in various ways. Corrected, new text lines 17- 18.
  2. In the Materials and Methods section, no information about the immunohistochemical assessment can be found. The authors must describe staining protocols, used antibodies etc. in detail, as well as their method of evaluating the slides. What is the expected localization of the staining -membranous, cytoplasmic or nuclear? Do positive/negative controls exist? If yes, please include images of both. If not, this has to be supplemented. We are remised that we omitted this important section. Detailed methodology is now included under section 4.2 as is the negative and positive controls for RPLP1 staining (Revised Figure 1).
  3. In the Results section (part 2.1) the authors speak of "minimal expression" and "little to no expression" of RPLP1 - how are these defined and how do they differ? Also, how is a significantly higher" expression of RPLP1 in secretory stage biopsies defined? The histologic pictures in Fig. 1 and Fig. 2 are rather small and especially images A and B in Fig. 1, as well as image A in Fig. 2 have a very blueish tone to them. They need to be improved and exchanged.

We have now clarified under section 4.2, how minimal, moderate and strong staining is defined including the scoring system (Fig. 8). Within the text, when we say significant, it is defined as a p-value <0.05 for each specified comparison. We have modified the figures as best we can.  We were given a 10 day time frame for resubmission which prevented us from further new IHC experimentations.

  1. How was the menstrual cycle stage determined? By histology alone or based on clinical information? Please clarify. The description “Stage of menstrual cycle was determined by the date of the subject’s last menstrual period for pre-menopausal subjects” is now included in lines 612 - 614.
  2. In the legend to Fig. 1 there is a mistake in line 137: it must read "A-e-S = adenomyosis - eutopic - secretory". This has been corrected (line 139).
  3. The commercially obtained TMA with endometrial cancer tissue might have completely different pre-analytic conditions (formalin concentration, time of fixation etc.) and therefore staining of the TMA specimens and the self-obtained tissue samples cannot simply be compared. Are there on-slide controls of normal/adenomyotic tissue within the commercial TMA? If yes, the authors should provide images and if not, this should be addressed in the Discussion. Also, the comparatively low numbers of patients with adenomyosis and endometriosis should be discussed critically. We thank you for this comment. We have checked with the provider of the tissue array (information is on their website) and the processing (formalin concentration, fixing time, etc. is identical to that used by our Pathology department who provides the inhouse specimens)  This information no appears in lines 610 - 612.
  4. Due to the higher density of epithelial tumor cells in the endometrial cancer specimens, a more intense staining is to be expected. Has this been accounted for in the assessment and how? This should also be added to the Materials and Methods section, as well as the Discussion. Thank you, we now include more detail on how the assessment was made as well as make mention in the Discussion section.
  5. In lines 220-222, the AN3-CA cell line is stated as having the highest levels of RPLP1 transcript expression, yet this cannot be deducted from Fig. 5, where AN3 shows the second lowest transcript expression. This is to be clarified. We have modified the Figure to make this more clear.
  6. As a whole, the concept of different letters indicating statistical significance in most of the figures is rather confusing and misleading. The authors should think of a more concise way to point out significance and improve the respective figures. We have clarified how the analysis is made. We have been using this approach for over 25 years and it is standard practice for how we analyze data by one-way ANOVA within and between specific groups.
  7. Line 52 must read "these two pathologic tissues", as this refers to adenomyosis and endometriosis only, not cancer. Corrected as suggested.
  8. There are a number of typos and grammatical errors throughout the manuscript that have to be corrected: please see lines 55, 63, 158, 187, 216, 217, 221, 274, 301, 332, 343, 347, 420 and 522. Thank you for pointing these out, they have been corrected.

Reviewer 2 Report

Thank you for conducting this important  study trying to reveal the role of RPLP1  up-regulation in the endometrial confusing disorders, minor comments should be taken in comsideration:

-Grammar and structural revision is required (eg, “protein elongation as is expressed as a 1174 nucleotide variant”)  in addition to some sentences which are started with small letters

- I recommend discussing in more details the possible diagnostic tools, including immunohistochemistry, for RPLP1 and its potential role in routine pathological examination and prognosis of adenomyosis and extrapelvic endometriosis.

Author Response

ijms-2164710_Response to reviewers

     We would like to thank the reviewers for their helpful insights.  We have answered each of the  queries as outlined below (response is underlined).  We also indicate where the changes appear in the revised manuscript herein as well as in the marked-up version of the revised manuscript.

.

Reviewer 2

Thank you for conducting this important  study trying to reveal the role of RPLP1  up-regulation in the endometrial confusing disorders, minor comments should be taken in consideration:

  1. Grammar and structural revision is required (eg, “protein elongation as is expressed as a 1174 nucleotide variant”) in addition to some sentences which are started with small letters.  This has been corrected.

  1. I recommend discussing in more details the possible diagnostic tools, including immunohistochemistry, for RPLP1 and its potential role in routine pathological examination and prognosis of adenomyosis and extra-pelvic endometriosis. We have added a section in the discussion (lines 556 - 575).

Reviewer 3 Report

It is an important topic and I have found the manuscript suitable for consideration after certain changes. Here are my comments: 

Figures: From figures 3-7, the y-axis ranges overlapped with the axis caption. Please correct the figures and use the same-sized fonts in all the figures.  

Discussion: 

- Please write " two-dimensional (2D) in vitro studies...." (page 9, line 415), as it is not three-dimensional (3D) assay. I would suggest writing also something similar to this, "further three-dimensional (3D) assays are required to confirm using extracellular and tumor microenvironment matrices to mimic the in vivo tumor conditions, as the 2D assays do not mimic them". This article could be cited also to show usage of human tissue derived matrices would provide more realistic findings.

https://doi.org/10.1016/j.yexcr.2018.06.037

- Please write the strengths and limitations of the study. 

Author Response

ijms-2164710_Response to reviewers

     We would like to thank the reviewers for their helpful insights.  We have answered each of the  queries as outlined below (response is underlined).  We also indicate where the changes appear in the revised manuscript herein as well as in the marked-up version of the revised manuscript.

Reviewer 3

It is an important topic and I have found the manuscript suitable for consideration after certain changes. Here are my comments: 

  1. Figures:From figures 3-7, the y-axis ranges overlapped with the axis caption. Please correct the figures and use the same-sized fonts in all the figures.  This has been corrected.
  2. Discussion: 

- Please write " two-dimensional (2D) in vitro studies...." (page 9, line 415), as it is not three-dimensional (3D) assay. I would suggest writing also something similar to this, "further three-dimensional (3D) assays are required to confirm using extracellular and tumor microenvironment matrices to mimic the in vivo tumor conditions, as the 2D assays do not mimic them". This article could be cited also to show usage of human tissue derived matrices would provide more realistic findings.  We have now included this information (lines 583 - 586) and site the study by Apu et al., as suggested.

https://doi.org/10.1016/j.yexcr.2018.06.037

Apu, E.-H.; Akram, S.U.; Fissanen, J.; Wan, H.; Salo T. Desmoglein 3 – influence on oral carcinoma cell migration and invasion.  Exp. Cell. Res. 2018, 370, 353 – 364.

  1. Please write the strengths and limitations of the study.  We have added a section on strengths and limitations (lines 576 - 590), thank you for the suggestion.

Round 2

Reviewer 1 Report

The authors have greatly improved their manuscript and all methods are now explained in detail.

There are yet several language errors/misspellings to be corrected (e.g. lines 231, 237, 240, 328, 459, 460). Also, a uniform way for the "H-Score" should be used throughout the paper. In addition, citation #52 is nowhere to be found in the text and should therefore be omitted.

Most notably, although the authors have included new images for their immunohistochemical findings, all the pictures of negative or weakly specimens still have a very blue tone to them compared to positive tissues and might be exchanged prior to publication to not raise the suspicion of undueful editing.

Author Response

We would like to thank reviewer for the additional suggestions which have enhanced the quality and clarity of the manuscript.  Concerns are addressed below.  We have answered each of the  queries as outlined below (response is underlined).  We also indicate where the changes appear in the revised manuscript herein as well as in the marked-up version of the revised manuscript indicated in the right hand column.

Reviewer 1

The authors have greatly improved their manuscript and all methods are now explained in detail.

  1. There are yet several language errors/misspellings to be corrected (e.g. lines 231, 237, 240, 328, 459, 460).

We apologize that we missed these errors in the last revision. We have spell and grammar checked the entire document again to correct any errors detected. 

  1. Also, a uniform way for the "H-Score" should be used throughout the paper. In addition, citation #52 is nowhere to be found in the text and should therefore be omitted.

We have changed H-Score to appear as such throughout the manuscript.

Reference #52 was corrected and remains in the references (we erroneously listed ref 50 twice in the text; reference #50 remains, and 51 and 52 replace where we had 50, 50 and 51 which now corresponds to 50, 51 and 52).

  1. Most notably, although the authors have included new images for their immunohistochemical findings, all the pictures of negative or weakly specimens still have a very blue tone to them compared to positive tissues and might be exchanged prior to publication to not raise the suspicion of undueful editing.

We apologize for the blue tone to the figures.  First, for clarification, we perform the IHC studies following the protocol recommended by Vector Laboratories when using the ABC kit from Vector.  The counterstain step is optional but we perform it for contrast between cell nucleus and DAB cytoplasmic staining for RPLP1 and we use hematoxylin (Zehntner et al., J. Histochem. Cytochem. 2008 Oct; 56(10): 873–880) which was purchased from Vector laboratories (H-3404-100).  Variation among samples (varying degree of blue tone) may be lot specific but we are uncertain.  IHC for the photomicrographs in Fig. 1 were not all conducted at the same time and that may explain the difference in hematoxylin color (blue vs. more purple).  In this revised version, we present IHC data for which all samples were run for IHC concurrently which gives a more uniform appearance for hematoxylin staining across all samples.  We have also included new positive and negative controls in this experiment using the same hematoxylin.  We specify the counterstaining as well in the revised manuscript and appears in lines 469 – 470.
